# Profiling the miRNA from Exosomes of Non-Pigmented Ciliary Epithelium-Derived Identifies Key Gene Targets Relevant to Primary Open-Angle Glaucoma

**DOI:** 10.3390/antiox12020405

**Published:** 2023-02-07

**Authors:** Padmanabhan Paranji Pattabiraman, Valeria Feinstein, Elie Beit-Yannai

**Affiliations:** 1Glick Eye Institute, Department of Ophthalmology, Indiana University School of Medicine, 1160 West Michigan Street, Indianapolis, IN 46202-5209, USA; 2Clinical Biochemistry and Pharmacology Department, Ben-Gurion University of the Negev, Beer-Sheva 84105, Israel

**Keywords:** oxidative stress, exosomes, miRNA, non-pigmented ciliary epithelium, trabecular meshwork, primary open-angle glaucoma

## Abstract

Oxidative stress (OS) on tissues is a major pathological insult leading to elevated intraocular pressure (IOP) and primary open-angle glaucoma (POAG). Aqueous humor (AH) produced by the non-pigmentary ciliary epithelium (NPCE) drains out via the trabecular meshwork (TM) outflow pathway in the anterior chamber. The exosomes are major constituents of AH, and exosomes can modulate the signaling events, as well as the responses of their target TM tissue. Despite the presence of molecular mechanisms to negate OS, oxidative damage directly, as well as indirectly, influences TM health, AH drainage, and IOP. We proposed that the expression of microRNA (miRNAs) carried by exosomes in the AH can be affected by OS, and this can modulate the pathways in target cells. To assess this, we subjected NPCE to acute and chronic OS (A-OS and C-OS), enriched miRNAs, performed miRNA microarray chip analyses, and miRNA-based gene targeting pathway prediction analysis. We found that various miRNA families, including miR27, miR199, miR23, miR130b, and miR200, changed significantly. Based on pathway prediction analysis, we found that these miRNAs can regulate the genes including *Nrf2*, *Keap1*, *GSK3B*, and serine/threonine-protein phosphatase2A (*PP2A*). We propose that OS on the NPCE exosomal miRNA cargo can modulate the functionality of the TM tissue.

## 1. Introduction

Oxidative stress (OS) is defined as an imbalance between the production and accumulation of reactive species in cells. Cells and tissues possess the ability to detoxify these reactive products [1]. In fact, OS has been proven to be a key factor in biological regulation under normal and pathological conditions [2]. The initial cause of OS is diverse and results in a series of common cellular events, which include modifications to protein, lipid, and nucleotide that can be tracked and measured [3]. Upon oxidative insult, the cellular defense mechanisms are activated, including increased expression of antioxidant enzymes, such as superoxide dismutase (SOD), catalase, and glutathione peroxidase (GPX), as well as augmenting their activity. In parallel, there is an increased consumption of low molecular weight antioxidants that includes ascorbic acid, glutathione, and uric acid [4]. A basal level of reactive oxygen species is essential to maintain various biological processes, including cell proliferation and differentiation [5]. When the degree of OS crosses a certain level, cellular and tissue damage responses occur. Several cellular signaling events occur in response to OS, including MAPK, PI3K, p53, Notch1, and nuclear factor erythroid 2–related factor 2/Kelch-like ECH-associated protein 1 (Nrf2-Keap1) signaling pathways [6]. These pathways have a rate, strength, and duration that vary depending on the source of the OS, and the number of cells and tissue exposed to it. The Nrf2/Keap1 pathway stands out among the pathways in its centrality, having the ability to regulate defense mechanisms against OS. The general regulators of Nrf2/Keap1 are detailed in many publications describing its role in different tissues and diseases [7,8,9]

Extracellular Vesicles (EVs) comprise a range of lipid-bound sacs with different properties and can be divided according to their cellular origin (cytosol or membrane), size (30 nm to 1 μm), and macromolecular contents [10]. The three major EV groups are exosomes, microvesicles, and apoptotic bodies [11]. Exosomes are a nano-scale (30–180 nm) cup-shaped subgroup of EVs, having a lipid bilayer membrane originating from cytoplasmic and specifically from multivesicular bodies (MVBs), mostly from all cell types. The ESCRT (endosomal sorting complexes required for transport) is the dominant machinery responsible for controlling sorted protein into the MVB. Exosomes are released into the extracellular space when the MVBs fuse with the plasma membrane [12]. Exosomes cargo includes miRNA, mRNA, ncRNA, ssDNA, and cytoplasmic and membrane proteins, thus playing a role in cell-to-cell communication. Exosomes, besides having a local effect on the same or neighboring tissues, can spread from their originating tissue via the bloodstream, lymph system, and extracellular fluids, reaching their distal targets [13]. The ability of exosomes to cross biological barriers due to their native lipophilic membrane allows them to diffuse passively through the BBB, placenta, and fat tissue [14]. Specific recognition of exosome membrane surface proteins and uptake mechanisms suggest the targeted delivery of messages by exosomes bring about physiological homeostasis and potentially function under pathological conditions [15].

Reactive oxygen species play a key role in the pathogenesis of primary open-angle glaucoma (POAG), a chronic optic neuropathy. It predominantly starts within the trabecular meshwork (TM) tissue impairment to sense the increase in the intraocular pressure (IOP). This IOP elevation induces apoptosis of the retinal ganglion cells (RGC), leading to blindness [16]. The AH produced by NPCE has a unique short one-way flow. Much of the AH is drained by the TM outflow pathway, including the TM, juxtacanalicular tissue, and, finally, via the Schlemm’s canal. The IOP is regulated primarily by fluid resistance to AH outflow. The extracellular matrix (ECM) of the TM is thought to be important in the regulation of IOP [17]. The TM tissue in the ocular drainage system is unique by being continuously exposed to OS [18]. However, we speculate that this exposure can vary the effects, due to either acute OS or chronic OS developed during elevated IOP. Both chronic and acute OS may result in cellular adaptations elicited by the exosome-mediated signals. In the present study, we will examine the changes taking place in NPCE-derived exosomes exposed to acute or chronic OS, further establishing its relevance in OS response genes. The AH carries functional EVs originating from all tissues making up the outflow pathway [19]. Our laboratory recently showed that EVs derived from oxidized NPCE cells protected the TM cells from the direct OS by significantly inducing Nrf2 [20]. In that study, we showed that treatment with EVs released by oxidatively stressed NPCE cells resulted in an increase in the Nrf2 staining in the TM cytoplasm and nucleus and induced Nrf2 protein levels significantly. Contrarily, EVs extracted from the NPCE cells unexposed to OS did not induce Nrf2 changes in the treated TM cells. The TM cells directly exposed to OS showed a significant change in Nrf2. Further, the downstream response included an increase in antioxidant genes and protein expression, including SOD1, SOD2, GPX1, heme oxygenase 1 (HMOX1), Nrf2, and increased catalase and SOD activities [20]. These suggest a strong link in the modulation of OS signaling in TM by EVs from NPCE. Moreover, recently mesenchymal stem cell-derived exosomes were shown to protect TM from OS [21]. The canonical Wnt pathway plays a major role in regulating AH drainage homeostasis. Key proteins involved in this pathway are pGSK, TGFβ, the PP2A phosphatase, and MMPs. Interestingly, these proteins are involved in controlling the activity of Nrf2/Keap1. Taken together, we predict that the major OS pathway, the Nrf2/Keap1, plays an important role in exosome-mediated OS cell responses in TM. This can potentially contribute to changes in the TM affecting IOP and participating in POAG pathology. Thus, the biology of OS and exosomes have merged our understanding of the paracrine function of exosomes, as well as the cellular responses by NPCE and TM to OS challenges [22]. As mentioned earlier, miRNAs are carried by the EVs, and such miRNAs can regulate the Nrf2 antioxidant pathway [23]. The miRNAs are involved in the regulation of various cellular processes in all biological tissues including the ocular tissues. Although there has been no single miRNA or a family that has been solely indicated in the pathogenesis of elevated IOP and POAG, there are multiple studies indicating the role of miRNAs in TM regulated by OS and supplementation of the miRNAs reversing OS, thus favoring a significant role of miRNA in modifying the oxidative damage to TM [24]. Additionally, the involvement of miRNAs in POAG has been investigated using models of mechanical stress, hypoxia, inflammation, apoptosis, and a combination of these models [25,26]. A set of miRNAs were indicated to participate in different signaling pathways known to affect TM cells, including, but not limited to the involvement of mTOR, MEK/ERK, TGF-β, PI3K/AKT, Wnt/β-Catenin, MMP-9 and Nrf2 [27]. However, very little information exists on the miRNAs regulated by OS in the NPCE and their resultant signaling changes on their target tissue, such as the TM. The present study aims to identify miRNA carried as exosomal cargo with the potential to affect the Nrf2/Keap1 pathway and relevant genes important in POAG pathogenesis when the NPCE tissue is exposed to OS.

## 2. Material and Methods

### 2.1. Cell Line

The human NPCE cell line was kindly supplied by Prof. Miguel Coca-Prados, Yale University [28]. The NPCE cell lines authentication test was performed at the Genomics Center of Biomedical Core Facility, Technion, Israel, using the Promega GenePrint 24 System. The Authentication report can be found in the Appendix A.

### 2.2. Oxidative Stress

To induce OS in NPCE cells, we utilized 2,2′-Azobis (2-amidinopropane) dihydrochloride (AAPH), which is a free radical-generating azo compound. AAPH is capable of initiating oxidation reactions by continuous production of peroxyl radical followed by alkoxyl radical via nucleophilic and free radical mechanisms [29]. NPCE cells were exposed to an acute OS 15 mM AAPH for 90 min for acute OS (A-OS) or 1.5 mM AAPH for 24 h as chronic OS (C-OS) stimulation.

### 2.3. Exosomes Isolation

Exosome-depleted media was prepared by first mixing Dulbecco’s modified Eagle’s medium with regular fetal bovine serum (FBS). This media was subjected to ultracentrifugation for 16 h at 110,000× *g* overnight at 4 °C using an SW28 rotor. Further filtration through a 0.22 µm filter (Millipore Express PLUS (PES) membrane) was carried out to completely deplete the exosomes. This media was used for downstream experiments. Upon conclusion of the experiments, exosomes secreted into the condition media (CM) from the cultured cells were prepared. Exosomes were isolated from CM of cultured NPCE cell line using a series of ultracentrifugation steps. At first, centrifugation at 300× *g* for 10 min to get rid of the cells was performed. Then, 2000× *g* for 10 min to get rid of the dead cells, followed by centrifugation at 10,000× *g* for 30 min to get rid of the cell debris. The final steps were at 100,000× *g* for 70 min, repeated twice, as detailed previously [30].

### 2.4. Exosomes Size and Concentration Analysis

Isolated exosomes were analyzed using the NanoSight NS500 instrument (Malvern Panalytical, Malvern UK) equipped with a blue laser (405 nm). Nanoparticles were illuminated by the laser and their movement under Brownian motion was captured for 60 s. Videos were then subjected to nanoparticle tracking analysis (NTA) using the NanoSight particle tracking software (NTA 2.0). At least three videos were captured for each sample to provide a representative concentration measurement, and all analysis settings were kept constant within each experiment. The settings were: camera level of 16–17; automatic functions for all post-acquisition settings; and camera focus were adjusted to make the particles appear as sharp dots. Using the script control function, three 30 s videos for each sample were recorded. Size distribution profiles obtained from NTA were averaged within each sample across the video replicates and then averaged across samples to provide representative size distribution profiles. These distribution profiles were then normalized to total nanoparticle concentrations or final cell counts. The data obtained by the NanoSight particle tracking software (NTA 2.0) is presented as mean exosome size and the most common exosome size (Mode). All experiments were performed at 1:1000 dilutions, yielding particle concentrations of ≈6 × 10^7^/mL.

### 2.5. Exosomes miRNA Extraction and Characterization

Total RNA extraction from exosomes was executed with a Total exosomal RNA purification kit (cat: 17200, Norgen Biotek Corp. Thorold, Ontario, Canada), as recommended by the manufacturer. In brief, 200 µL of exosomes in PBS were lysed for 1 min in lysis buffer by vortexing. Spin columns were used for miRNA separation at 3500× *g* for 1 min. Columns were washed four times at 14,000× *g* for 1 min, following elution in 50 µL, at 600× *g* for 2 min. miRNA quality and integrity were assessed with Bioanalyzer 2100 (Agilent Technologies, Santa Clara, CA, USA). Samples with mRNA concentrations higher than 130 ng/µL were further analyzed. Exosomal miRNA were characterized with the suitable microarray chip analysis GeneChip^®^ miRNA 4.0 Array and Flashtag™ Bundle–ThermoFisher (cat 902445).

### 2.6. miRNA-Based Gene Targeting Pathway Prediction Analysis

We collected information about miRNAs that were reported or predicted to target nuclear factor erythroid-derived 2-like 2 (NFE2L2) or Nrf2 from the following resources: miRDB [31], high-throughput datasets generated with cross-linking ligation and sequencing of hybrids (CLASH) method [32], TarBase [33], and miRTarBase [34]. miRNA expression array files (CEL) were uploaded into Transcriptome Analysis Console (TAC) software v4.0.3 to identify miRNAs that are differentially expressed in OS versus control samples (Fold-change (OS vs. control) < −2, p-adj < 0.1).

Lists of *NFE2L2*, *Keap1*, *GSK3B*, and *PPP2CA* targeting miRNAs were intersected with differentially expressed miRNAs to identify miRNAs whose decrease in exosomes could be directly linked to the activation of *NFE2L2*, *Keap1*, *GSK3B*, and *PPP2CA*.

### 2.7. Statistical Analysis

Raw data were extracted automatically in Affymetrix data extraction protocol using the software provided by Affymetrix GeneChip^®^ Command Console^®^ 4.3.2 (AGCC) Software. The CEL files were imported and changes in miRNA levels were performed using Affymetrix^®^ Expression Console™ Software. Array data were filtered by probes annotated species. Comparative analysis was carried out between test and control samples using fold-change and independent T-test, in which the null hypothesis was that no difference exists between the two groups. The false discovery rate (FDR) was controlled by adjusting the *p*-value using the Benjamini–Hochberg algorithm.

## 3. Results

### 3.1. NPCE Isolated Exosomes Characterization

Extracted NPCE-derived exosomes following acute and chronic OS were compared to the NPCE exosomes derived from the control treatment for differences in size and concentration using NTA. The A-OS or C-OS exposure on the NPCE cells did not affect the exosome size or concentration compared to the control or between treatments. The mean size was found to be 149.1 to 150 nm. There was no significant change in the mode, suggesting that the exosomes released were not different in size (Table 1 and Figure 1). The data represents the average of n = 3 (Table 1).

### 3.2. miRNA Predicted Nrf2 Gene Target Analysis

Four databases were used for *Nrf2* gene target prediction. TarBase analysis suggested 96 potential miRNAs; miRDB analysis suggested 64 potential miRNAs and miRTarBase analysis revealed 9 miRNAs; whereas the CLASH data could not predict any miRNA predicted for the *Nrf2* gene.

Among all the potential miRNAs predicted to affect the *Nrf2* gene, three miRNAs—hsa-miR27b-3p, hsa-miR199a-5p, and hsa-miR199a-3p—were found in the three data-bases. Moreover, these three miRNAs were found following A-OS and C-OS. A significant increase in signal from the NPCE-derived exosomes of hsa-miR27b-3p was found in A-OS exposed NPCE cells from control and C-OS (Figure 2).

The hsa-miR199a-5p and hsa-miR199a-3p levels were not different from the control following A-OS. However, following C-OS there was a significant decrease in the miRNA levels compared to control, as well as A-OS (Figure 2).

### 3.3. miRNA Predicted GSK3B Gene Target Analysis

Among the proteins that control the phosphorylation of Nrf2 is GSK3B [35]. Therefore, we looked for miRNAs present in the NPCE-derived exosomes, which can modify *GSK3B* expression that is differentially expressed under OS versus control samples.

The hsa-miR199a-3p, which was predicted to modify the *Nrf2* gene, can also modify the *GSK* gene expression. There is a significant signal increase following A-OS compared to control and C-OS. A similar pattern was found for hsa-miR27a-3p, whose signal is significantly higher after A-OS compared to control and C-OS. However, the degree of hsa-miR27a-3p signal in control and under OS was 50% further decreased than hsa-miR-199a-3p. The hsa-miR24-3p signal was significantly higher in A-OS versus C-OS, with a relatively high signals in control and OS group relative to the other miRNA-predicted *GSK3B* gene targets analyzed. The hsa-miR23b-3p and hsa-miR29b-3p were predicted *GSK3B* gene targets without significant differences between control, C-OS, and A-OS (Figure 3).

### 3.4. miRNA Predicted PPP2CA Gene Target Analysis

The interrelation and regulation between PP2A and Nrf2 proteins are associated with OS [36]. We examined miRNAs presented in the NPCE-derived exosomes that are differentially expressed under OS versus control samples, which we believe can modify *PPP2CA* expression. All three detected miRNAs—hsa-miR197-3p, hsa-miR130b-3p, and hsa-miR125-5p—showed an increased signal after A-OS compared to C-OS. This response was particularly noticeable for hsa-miR197-3p, with a significant signal increase compared to the control (Figure 4).

### 3.5. miRNA Predicted Keap1 Gene Target Analysis

Nrf2 in the cytoplasm binds to Keap1, which, in turn, facilitates the ubiquitination and subsequent proteolysis of Nrf2 [37]. Upon oxidative stress, the Nrf2-Keap1 complex depredates and Nrf2 can translocate to the nucleus and regulates the cellular resistance to oxidants. We looked for miRNAs presented in the NPCE-derived exosomes that are differentially expressed under OS versus control samples, as this can modify *Keap1* expression. Only one miRNA, hsa-miR200a-3p, came up in three databases: miRDB, data TarBase, and miRTarBase. Under C-OS, a trend of reduction in the hsa-miR200a-3p signal was found, with a further trend of reduction under A-OS (Figure 5).

## 4. Discussion

Here, we have taken a systems biology approach to successfully examine the contributions of OS on the changes in NPCE exosomal miRNAs. Further, we have predicted how these changes in miRNAs can affect the key genes related to the OS response pathway: *Nrf2*, *Keap1*, *PPP2CA*, and *GSK3B*.

Our experiments comparing NPCE exosome size following A-OS or C-OS followed the general acceptance of lack of size changes in EVs under stress. In general, we were curious to see if OS affected exosome size and/or concentration. Earlier, this phenomenon was addressed by F.J Romero’s lab, which reported that stressed retinal pigment epithelium (ARPE-19) cells released a higher number of exosomes compared to the controls without significantly changing the exosome size [38]. In another study, thermal stress and OS increased the release of exosomes by Jurkat and Raji cells without affecting the exosome size [39]. Additionally, the stressors, including hypoxia, TNF-α-induced activation, and high glucose, did not affect exosome size [40]. In contrast, osmotic stress was shown to increase the size and enhance the release of exosomes derived from fibroblasts. Today, it is accepted that some OS might trigger an increase in exosome release, with evidence for OS influencing the exosome cargo loading [22]. Interestingly, there is no evidence of exosomal size changes due to OS.

Our lab has been studying naïve exosomes derived from NPCE cells and their effects on modulating the canonical Wnt signaling pathway and ECM remodeling [41,42,43,44]. Earlier, we were able to show the involvement of key proteins in the canonical Wnt pathway and the PDK1/Akt/mTOR signaling pathways [20]. Additionally, we found that the phosphatase PP2A, known to catalyze the dephosphorylation of Akt, was activated in TM cells upon treatment with NPCE-derived exosomes [41]. PP2A aids in the partial deactivation of Akt and activation of GSK3β by dephosphorylating [20]. These proteins were found to be related to control Nrf2 and to be involved in the Wnt pathway, which are important in the pathogenesis of POAG [45,46]. In the current study, enhancement of the miR27a signal was found following A-OS. Increased miR27a expression was found following *in vitro* human TM cells exposed to OS, resulting in the activation of PI3K/AKT and Wnt/β-catenin pathways [47]. The NPCE-derived exosomal miR199a-3p and miR199a-5p signals significantly decreased exclusively in C-OS compared to the control. This supports our understanding that miR199a-3p and miR199a-5p has a potential role to inhibit *Nrf2* expression under C-OS. Recently, Tian et al. showed a similar role of miR199b-3p in regulating the *Nrf2* pathway in a kidney injury model [48]. Interestingly, miR199-5p was suggested to be downregulated by OS-mediated TGFβ2 expression [49]. Elevated levels of TGFβ2 in the POAG patient’s AH, is a key growth factor known to be involved in the induction of both TM and optic nerve changes in POAG [50]. In POAG patients, the TM demonstrates altered actin cytoskeleton, ECM composition and this is accompanied by dysregulation of multiple signaling pathways, including but not limited to TGFβ [51] and Wnt/β-catenin, which is reduced under OS [52].

GSK3B is a key protein in the control and regulation of the Wnt and Nrf2 pathways [52]. The miRNAs regulating GSK3B expression and potentially its activity has a wide impact on cellular response against OS. This can extend into modulation of the extracellular matrix (ECM) remodeling in TM [53]. The ECM remodeling in TM plays an important role in the regulation of IOP [54]. For the first time, we were able to report the presence of members of two of the miR199 family predicted to affect GSK3B. The miR199a-5p signal changed significantly under OS compared to the control. Interestingly, miR199 was not found in the POAG human AH exosomes in the earlier report [19,55], but was detected in AH from keratoconus eyes [56]. The significance of the NPCE-derived exosomes on TM among other exosome sources within anterior segment that gets into the AH needs further investigation.

miR29b is involved in the regulation of ECM by controlling the synthesis and deposition of ECM proteins in TM cells [57]. Chronic OS can significantly decrease miR29b and an induction in the expression of miR29b has been shown to upregulate ECM genes, including collagens, laminin, and fibrillin. In our study, for the first time, we show that hsa-miR27a-3p is present in NPCE-derived exosomes and that its signal significantly increased under A-OS. This helps supporting the hypothesis that miR27a-3p can induce modulate Wnt pathway. This is based on recently published data which suggests that miR27a-3p modulates the Wnt pathway via the sFRP-1 antagonist [58].

The Serine/threonine-protein phosphatase 2A is encoded by the *PPP2CA* gene and is reported to be significantly higher in AH samples from the POAG patients relative to cataract controls [59]. PP2A has a role in the modulation of GSK and negatively regulates Wnt signaling networks [20]. In addition, PP2A activity is reduced following OS via AKT signaling. This makes PP2A an important hub between the OS and the Wnt pathway. In the present study hsa-miR197-3, hsa-miR130b and hsa-miR125-5p, which are predicted to modulate the *PPP2CA* gene, were found in NPCE-derived exosomes and their signals significantly increased following A-OS. Limited data exist regarding the role of these miRNAs in the ocular drainage system. which we plan to investigate in the future.

The hsa-miR200a-3p was the only miRNA predicted by three databases to affect the *Keap1* gene. Through the ability to regulate *Nrf2* activation by targeting Keap1, the miR-200a-3p may play a critical role in modulating cellular anti-oxidant signals. Evidence for its relevance in the ocular drainage system can be found in miR200a upregulation, which has been shown to positively affect retinal ganglion cell pathologies [60].

The major limitation of our study is the lack of experimental target validation. Further experimental evidence will help us decipher, in greater detail, the effects of these oxidatively stressed exosomes on changes in biological pathways in the target cells/tissue in question, which is the TM.

To summarize, the present research contributes another layer to the understanding of how NPCE-derived exosomes can potentially affect the TM, while focusing on miRNAs that can regulate the activity of genes central to the pathophysiology of glaucoma.

## Figures and Tables

**Figure 1 antioxidants-12-00405-f001:**
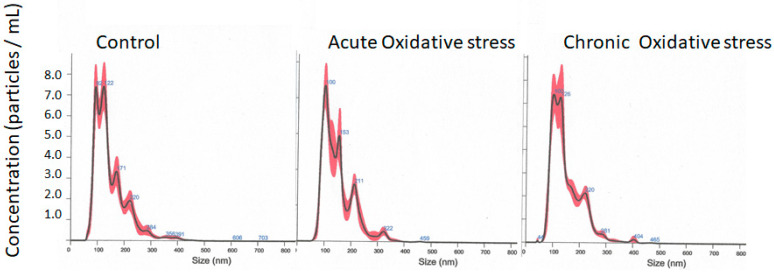
Nanoparticle tracking analysis (NTA) of NPCE exosomes isolated by ultracentrifugation. Data represent the average size distribution profile of n = 3 for each treatment derived from three different videos and analyses.

**Figure 2 antioxidants-12-00405-f002:**
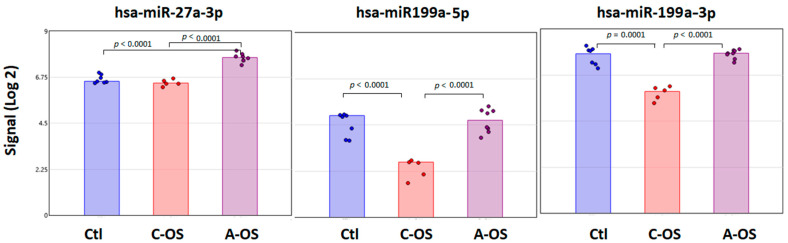
Individual changes in expression of exosomal miRNAs predicted to target the *Nrf2*. Bar plots depict the Affymetrix signal of hsa-miR27a-3p, hsa-miR199a-5p, and hsa-miR199a-3p in NPCE cells-derived exosome following C-OS or A-OS. Each group contains a minimum of five independent replicates. One-way ANOVA was performed to identify the *p* < 0.001 among these three groups.

**Figure 3 antioxidants-12-00405-f003:**
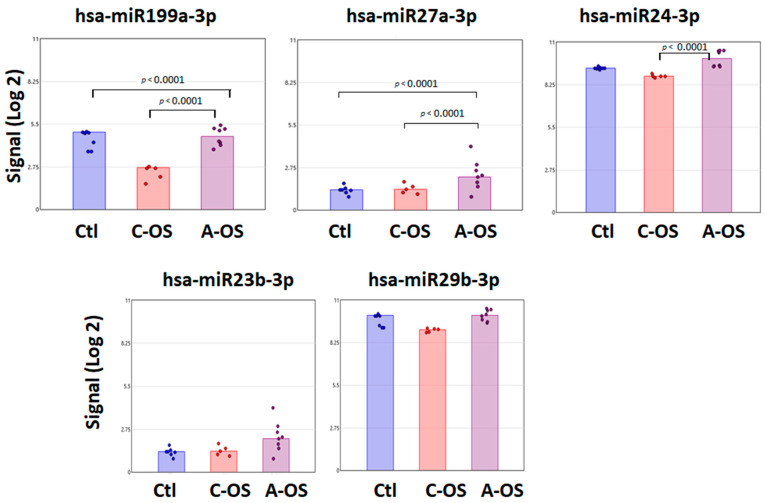
Individual changes in expression of exosomal miRNAs predicted to target *GSK*. Bar plots depict the Affymetrix signal of hsa-miR199a-3p, hsa-miR27a-3p, hsa-miR24-3p, hsa-miR23b-3p, and hsa-miR29b-3p in NPCE cells-derived exosome following C-OS or A-OS. Each group contains a minimum of five independent replicates. One-way ANOVA was performed to identify the *p* < 0.001 among these three groups.

**Figure 4 antioxidants-12-00405-f004:**
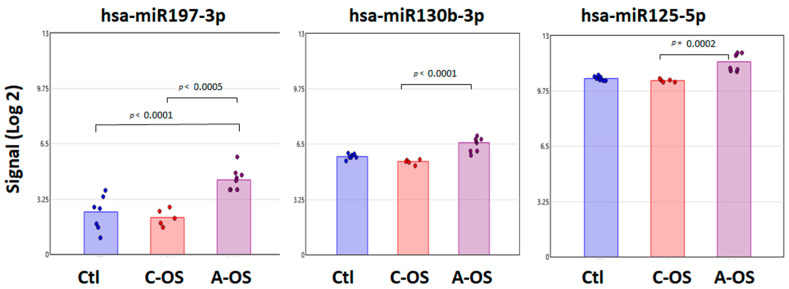
Individual changes in expression of exosomal miRNAs predicted to target *PPP2CA*. Bar plots depict the Affymetrix signal of hsa-miR197a-3p, hsa-miR130b-3p, and hsa-miR125b-5p in NPCE cells derived-exosome following C-OS or A-OS. Each group contains a minimum of five independent replicates. One-way ANOVA was performed to identify the *p* < 0.001 among to these three groups.

**Figure 5 antioxidants-12-00405-f005:**
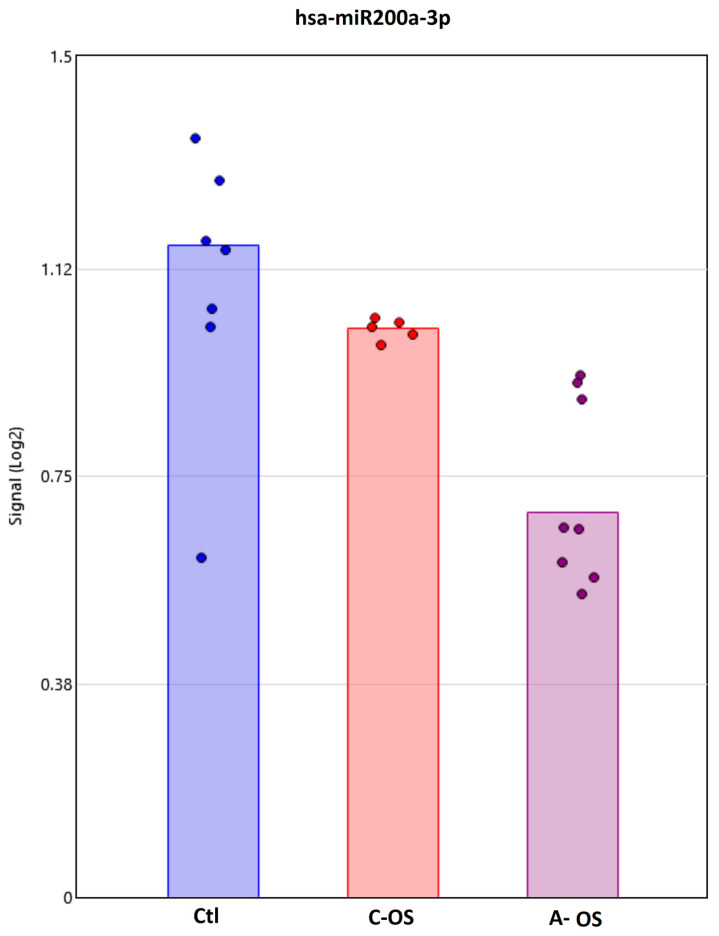
Individual changes in expression of exosomal miRNAs predicted to target *Keap1*. Bar plots depict the Affymetrix signal of hsa-miR200a-3p in NPCE cells-derived exosome following C-OS or A-OS. Each group contains a minimum of five independent replicates. One-way ANOVA was performed to identify the *p* < 0.001 among these three groups.

**Table 1 antioxidants-12-00405-t001:** Characterization comparison of NPCE-derived exosomes following acute and chronic oxidative stress.

	Mean (nm)	Mode (nm)	Concentration (Particles/mL)
**Control**	149.1 ± 6.3	97.5 ± 5.3	2.34 × 10^7^ ± 6.80 × 10^8^
**Acute Oxidative stress**	149.6 ± 1.8	103.8 ± 6.2	3.04 × 10^7^ ± 7.30 × 10^8^
**Chronic Oxidative stress**	150.0 ± 2.4	115.6 ± 6.8	4.99 × 10^7^ ± 6.04 × 10^8^

## Data Availability

The dataset published in this manuscript is part of a larger dataset that has not been shared onto data repository. Requests for material can be made to the corresponding authors.

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
