# Peer review of "Profiling the miRNA from Exosomes of Non-Pigmented Ciliary Epithelium-Derived Identifies Key Gene Targets Relevant to Primary Open-Angle Glaucoma"

_antioxidants, 2023, doi:10.3390/antiox12020405_

Round 1

Reviewer 1 Report

Authors aimed to identify miRNA carried as cargo of exosomes with the  potential to affect the Nrf2/Keap1 pathway and relevant gene important in POAG when  parental cells exposed to oxidative stress. They added the oxidative stress to the Human NPCE cell line ain vitro and extract the exosome to measure the cargo contents.

Overall the study is well designed and supported by their data. The only limitation is that these findings need verrification in an in vivo experiments.

Minor concerns: Line156: Is hsa-miR-199b-3p a typo error of hsa-miR-199a-5p?

Author Response

We thank all the reviewers for their helpful and encouraging comments. We have addressed all the comments to the best of our ability.

Reviewer 1

Overall the study is well-designed and supported by their data. The only limitation is that these findings need verification in vivo experiments.

We completely agree with the reviewer and we have clearly stated that the major limitation is the lack of experimental target validation which is the TM (Line 354).

Minor concerns: Line156: Is hsa-miR-199b-3p a typo error of hsa-miR-199a-5p?

We would like to thank the reviewer for pointing out this mistake. We have corrected the relevant paragraph and Figure 2 (in the original submission it was Figure 1)

Among all the potential miRNAs predicted to affect the Nrf2 gene, three miRNAs: hsa-miR-27b-3p, hsa-miR-199a-5p, and hsa-miR-199a-3p were found in the three databases. Moreover, these three miRNAs were found following A-OS and C-OS. A significant increase signal in the NPCE-derived exosomes of hsa-miR-27b-3p was found in AOS ex-posed NPCE cells vs. control and C-OS (Figure 2). " The hsa-miR-199a-5p and hsa-miR-199a-3p levels were not different from the control following A-OS. However, following C-OS there was a significant decrease in the miRNA levels compared to control as well as A-OS (Figure 2) "

Reviewer 2 Report

Pattabiraman and colleagues report on the differential expression of multiple miRNAs with a potential impact on genes that play a role in the pathogenesis of open-angle glaucoma. The findings add useful information to the current knowledge and can inspire further research in this field. The manuscript has been organized and written very well. The findings are presented and discussed appropriately. I have a few minor comments for the authors:

1. The research methods and findings do not support the conclusion regarding the trabecular meshwork function as it has not been tested here. Hence, I would suggest revising the conclusion in the abstract and the main manuscript accordingly.

2. Please mark significant differences in the remaining plots of Figure 2 as well as Figure 4. 

3. The use of acronyms should be consistent. For example, "ROS" has been defined in lines 39 and 65. 

4. The use of punctuation needs a thorough check. For example, in line 38: a comma is required between ascorbic acid, glutathione, uric acid, ... Also, in line 52: remove the extra "(." and in line 192: (Figure 3).

5. Line 191: "...hsa-miR-197-3p who significantly signal increase ..." does not make sense. Please consider revising.

Author Response

We thank all the reviewers for their helpful and encouraging comments. We have addressed all the comments to the best of our ability.

  1. The research methods and findings do not support the conclusion regarding the trabecular meshwork function as it has not been tested here. Hence, I would suggest revising the conclusion in the abstract and the main manuscript accordingly.

The conclusions are rephrased as:

Line 24 (Abstract): "We propose that OS on the NPCE exosomal miRNA cargo can modulate the functionality of the TM tissue."

Line 354 (Discussion): We clearly state that the research's major limitation is the lack of experimental target validation which is the TM.

Line 358: we summarized " the present research contributes another layer to the understanding of how NPCE-derived exosomes can potentially affect the TM

  1. Please mark significant differences in the remaining plots of Figure 2 as well as Figure 4.

We have added Figure 1. So Figure 2 is now Figure 3.

Line 245-246:  "The hsa-miR23b-3p and hsa-miR29b-3p were predicted GSK3B gene target without significant differences between control, C-OS, and A-OS”

Figure 4 is now Figure 5

Significant differences were added to the Figure. If it is not significant, we did not represent the histograms as non-significant.

Reviewer 3 Report

Pattabiraman et al. realized a very interesting article describing the “Profiling the miRNA from exosomes of non-pigmented ciliary epithelium-derived identifies key gene targets relevant to primary open-angle glaucoma”. I consider the manuscript very interesting but, at the same time, I suggest several revisions needed to improve the reliability and the completeness of the paper: 

·      The “Introduction” sections should be more detailed and improved. I suggest adding data related to the involvement of oxidative stress and related pathways/genes/ncRNAs in the etiopathogenesis of open-angle glaucoma and possibly related components, especially with vascular ones. The recent PMID: 32877751, PMID: 30523548, PMID: 36490268 and PMID: 36290689 could represent a substrate able to enforce the role of considered cellular mechanisms.

·      Finally, manuscript requires important English revisions and typos correction. 

Author Response

We thank all the reviewers for their helpful and encouraging comments. We have addressed all the comments to the best of our ability.

The “Introduction” sections should be more detailed and improved. I suggest adding data related to the involvement of oxidative stress and related pathways/genes/ncRNAs in the etiopathogenesis of open-angle glaucoma and possibly related components, especially with vascular ones. The recent PMID: 32877751, PMID: 30523548, PMID: 36490268 and PMID: 36290689 could represent a substrate able to enforce the role of considered cellular mechanisms.

We have added more details in the introduction regarding the miRNA involvement in POAG, though there is no study showing any direct role in the etiology or pathogenesis. We have tried to add to the best of our knowledge the relevant publications, which resonate with the concern of the reviewer.

Finally, the manuscript requires important English revisions and typos correction. 

Intensive English revision and typos correction auditing were performed.